# Does Exposure to High Job Demands, Low Decision Authority, or Workplace Violence Mediate the Association between Employment in the Health and Social Care Industry and Register-Based Sickness Absence? A Longitudinal Study of a Swedish Cohort

**DOI:** 10.3390/ijerph19010053

**Published:** 2021-12-21

**Authors:** Anna Nyberg, Paraskevi Peristera, Susanna Toivanen, Gun Johansson

**Affiliations:** 1Department of Public Health and Caring Sciences, Uppsala University, P.O. Box 564, SE-751 22 Uppsala, Sweden; 2Institute of Environmental Medicine, Karolinska Institutet, SE-113 65 Stockholm, Sweden; Gun.johansson@ki.se; 3Department of Psychology, Stress Research Institute, Stockholm University, SE-106 91 Stockholm, Sweden; paraskevi.peristera@su.se; 4School of Health, Care and Social Welfare, Mälardalen University, P.O. Box 883, SE-721 23 Vasteras, Sweden; susanna.toivanen@mdh.se

**Keywords:** structural equation model, multilevel model, mediation model, indirect effect, industry level

## Abstract

Background: The aim of this paper was to investigate if job demands, decision authority, and workplace violence mediate the association between employment in the health and social care industry and register-based sickness absence. Methods: Participants from the Swedish Longitudinal Occupational Survey of Health who responded to questionnaires in 2006–2016 (*n* = 3951) were included. Multilevel autoregressive cross-lagged mediation models were fitted to the data. Employment in the health and social care industry at one time point was used as the predictor variable and register-based sickness absence >14 days as the outcome variable. Self-reported levels of job demands, decision authority, and exposure to workplace violence from the first time point were used as mediating variables. Results: The direct path between employment in the health and social care industry and sickness absence >14 days was, while adjusting for the reverse path, 0.032, *p* = 0.002. The indirect effect mediated by low decision authority was 0.002, *p* = 0.006 and the one mediated by exposure to workplace violence was 0.008, *p* = 0.002. High job demands were not found to mediate the association. Conclusion: Workplace violence and low decision authority may, to a small extent, mediate the association between employment in the health and social care industry and sickness absence.

## 1. Introduction

### 1.1. Sickness Absence in the Health and Social Care Industry

The health and social care industry is the largest industry in Sweden and employed about 16% of the workforce in 2020 (Statistics Sweden, www.scb.se, accessed on 18 October 2021). Compared with employees in many other industries, employees within health and social care have an increased risk of sickness absence in Sweden [1], as well as in the developed world at large [2,3,4]. The reasons for the increased risk in sickness absence are not fully understood, but poorer working conditions have been suggested to be a plausible contributing factor, particularly poorer psychosocial working conditions [1,4,5]. However, despite the fact that higher sickness absence rates among health and social care personnel are often assumed to be due to poorer working conditions in this industry, there are no empirical studies of this, to the best of our knowledge.

### 1.2. Job Demands and Decision Authority in the Health and Social Care Industry

Job demands and decision authority are two psychosocial work factors measured in the well-established demand-control model [6]. Job demands and decision authority have been reported by a representative sample of the Swedish working population, in the Swedish Work Environment Surveys (SWES), since 1989. In a study using data from SWES between 1991 and 2013, it was shown that the psychosocial work environment in the health and social care industry had deteriorated considerably in Sweden, with increases in high job demands and low decision authority [7,8]. These work factors did not only deteriorate over time but were also across time, in both genders, reported to be poorer in the health and social care industry than in most other industries. High job demands and poor decision authority are risk factors for a number of ill-health outcomes. For example, in systematic reviews, high job demands [9,10,11,12] and poor decision authority [9,12,13] are associated with increased risks of common mental disorders, which are the most common diagnoses behind sickness absences in Sweden [14]. High demands and poor job control or decision authority are also directly associated with increased risks of sickness absence in general working populations [15,16,17,18,19,20], specifically within health and social care [21,22,23].

### 1.3. Workplace Violence in the Health and Social Care Industry

Workplace violence is a workplace hazard that encompasses both psychological (e.g., bullying and harassment) and physical (e.g., kicks, hits, bites) behaviours, or threats of such behaviours [24]. Psychological forms of violence, primarily bullying, have been extensively studied, whereas the knowledge on health effects of physical violence or threats thereof, is somewhat more limited [25]. Physical violence is known to be considerably higher in the health and social care industry compared to others, and it is well established that the majority of violent events involve patients or clients. Personnel in emergency care, psychiatry, and elder care are particularly exposed [25,26]. The prevalence of workplace violence has, according to the Swedish Work Environment Survey, not increased across time in Sweden, but on the other hand it has remained at a stable, high level in the health and social care industry [8,25]. Associations between physical workplace violence, or threats thereof, and sickness absence have been found in a few studies, all from northern European countries [27,28,29]. It was, however, concluded in a recent systematic review that more longitudinal research is needed before conclusions about such an association in health and social care personnel can be drawn [30].

### 1.4. Variation in Sickness Absence and Psychosocial Work Factors on Industry Level

In a previous study based on the same cohort as the present one, exposure to workplace violence together with high emotional demands and poor work-time control, has been found to explain higher risk of self-reported sickness absence among human service employees, of which many work in the health and social care industry compared to other workers [31]. Also, other studies of specific health and social care occupations report similar results. For example, in a recent systematic review of nurses (the largest occupational group within health and social care) the authors found that high job demands increased the risk for sickness absence [32].

Health and social care is, to a large extent, organized and governed on the industry level and therefore the work environment is similar for many professions in this industry. Parts of the responsibility for the work environment in health and social care must be placed on the industry level, although differences in working conditions between professions also exist [7]. In the present study, we therefore investigate to what extent the work factors of job demands, decision authority, and workplace violence can mediate the higher levels of sickness absence rates in the health and social care industry compared to other industries. Thus, we investigate explanations for the differences in sickness absence on the *industry level* as opposed to the occupational one. The aim of the present study is, thus, to investigate if the psychosocial work factors of job demands, decision authority, and workplace violence mediate the association between employment in the health and social care industry and register-based sickness absence.

## 2. Materials and Methods

### 2.1. Study Design

This survey is based on a longitudinal open cohort with information from questionnaires at six waves between 2006 and 2016.

### 2.2. Study Sample

We used the Swedish Longitudinal Occupational Survey of Health (SLOSH), an open cohort representing an approximately nationally representative sample of the Swedish working population. The overall aim of the SLOSH survey is to facilitate studies of how psychosocial work factors, the interface between work and family life, and the transition to retirement, among others, affect health outcomes over time. By continuously inviting participants of the Swedish Work Environment Surveys (SWES) to follow-ups, Statistics Sweden have collected data from the SLOSH cohort every second year since 2006. Today SLOSH comprises SWES participants from 2003 until 2011 with a sample size of over 40,000 individuals. SWES consist of a subsample of gainfully employed people aged 16–64, stratified by county, sex, citizenship, and inferred employment status from the Labour Force Survey (LFS). SLOSH respondents are invited to answer a self-completion questionnaire in two versions, one for those who work 30% or more of full time and another one for those who work less than 30% or not at all. More information on SLOSH can be found in the cohort profile [33]. The SLOSH data from 2006 comprised 5985 individuals, from 2008 11,441 individuals, 2010 11,525 individuals, 2012 9880 individuals, 2014 20,316 individuals, and from 2016 19,360 individuals. The response rate has varied between 65.4% and 50.9% over these years. For the present study we selected participants who answered the questionnaire for those who work 30% or more in 2006 (*n* = 5141; 46.78% men, 53.22% women). Then we selected those that worked, at least, in two consecutive waves, resulting in *n* = 3964, which reduced to 3951 (46.22% men, 53.78% women) when excluding non-valid values for the covariates. Of these, 780 participants (19.92% of all; 13.46% of men, 86.54% of women) were employed in the health and social care industry in 2006 (wave 1). Information on industry and sickness absence was obtained from the Longitudinal Integration Database for Health Insurance and Labour Market Studies (LISA), which is an integrated total population register of all residents in Sweden 16 years and older, provided by Statistics Sweden. Register data on sickness absence in 2016 was linked to self-reported data provided by SLOSH participants. The participants received written information about the study and, in accordance with Swedish regulation and practice, responding to and returning the survey indicated informed consent. The Regional Research Ethics Board in Stockholm approved the study (Dnr: 2017/237-31). The funding sources had no role in the writing of the manuscript or in the decision to submit it for publication.

### 2.3. Variables

#### 2.3.1. Outcome Variable

##### Sickness Absence

Sickness absence was assessed as the total amount of register-based net days of sickness benefit 15 days or more/year, i.e., the number of days per year receiving sickness benefit from the Swedish Social Insurance Agency (SSIA). In Sweden, the employers pay for the first 14 days of sickness absence, which means that benefits from SSIA is paid from day 15 in the sickness absence period. These benefits can be given as 25%, 50%, 75%, or 100% of the working time. One net day of sickness benefit, which also includes preventive sickness benefit, rehabilitation allowance, or occupational injury allowance, can equal either one full day of sickness benefit, two days of 50% sickness benefit, or four days of 25% sickness benefit. We dichotomized information of net day of sickness benefits into those who had no days of sickness benefit in the year of the SLOSH data collection and those who had one or more days of sickness benefit the year of the study.

#### 2.3.2. Predictor Variable

Measurement of employment in the health and social care industry was based on the Swedish Standard Industrial Classification (SNI), a system that follows the recommendations of the statistical classification of economic activities in the European Community (NACE). The SNI classification system is an activity classification, with production units classified according to the main activity carried out. In this study, we created a dichotomous variable indicating (1) employment in the industry health and social care and (0) employment in any other sector.

#### 2.3.3. Mediating Variables

##### Psychosocial Work Factors

Job demands and Decision authority were measured by self-reports using the Demand Control Questionnaire (DCQ) [34,35], a widely used questionnaire operationalizing the demand–control–support model. Four items (working fast, too much effort, enough time, and conflicting demands, Cronbach’s alpha 0.66–0.68) were used to create a score for demands at work. Two items (deciding what to do at work, deciding how to do your work, Cronbach’s alpha 0.74–0.79) were used to create the variable decision authority at work. Workplace violence was measured with the question. “Were you exposed to violence or the threat of violence in your work the last six (or twelve) months?”. Between 2006 and 2010, the time period referred to was the last 12 months, whereas thereafter (2012–2016) the question referred to the time period of the last six months. The response alternatives were dichotomized to “no” or “yes”, where yes indicated being exposed to violence at least once during the period in question.

#### 2.3.4. Covariates

Age was adjusted for by the groups <35 years, 35–44, 45–54, 55–64, and >64 years old; gender was a dichotomous variable; marital status was adjusted for using the categories married/co-habiting or not, and having children living at home as a dichotomous variable with the response alternatives “yes” and “no”; and education has five categories ranging from one (less than nine years of education) to five (research education).

### 2.4. Analytical Strategy

#### 2.4.1. Descriptive Statistics

Descriptive statistics of the study population in 2006 were computed for the whole sample as well as separately for men and women.

#### 2.4.2. Autoregressive Cross-Lagged Mediation Models within a Multilevel Structural Equation Modelling (MSEM) Framework

Multilevel autoregressive cross-lagged mediation models (MSEM) were fitted to our data [36,37]. Since multiple measurement points (level 1) are nested within individuals (level 2) in our sample, a two-level SEM model that allows partitioning between- and within-person effects was implemented to account for two inherent types of heterogeneity, within people across time and between people [38,39,40]. The autoregressive cross-lagged models are among the most popular approaches in mediation analysis with longitudinal panel data [38] since they allow time for causes to have their effects, support stronger inference about the direction of causation in comparison, and reduce the probable parameter bias [37]. In this work, we simultaneously address the reciprocal temporal relationships between employment in the health and social care industry (HLTH, exposure variable), psychosocial work characteristics for high job demands (DEM), low decision authority (DA), workplace violence (VIOL) (putative mediators), and a sickness absence of 15 days or more (SA, outcome). Bivariate models were first fitted to examine whether there were cross-lagged relationships between the exposure of interest and the putative mediators, and between the putative mediators and outcome, which is a prerequisite for a causal pathway [41]. We tested models where high job demands and low decision authority were fitted either as latent variables with four and two items or as observed variables (considering the mean of the various items) and finally presented only the results for the models with the best fit. The fit of the measurement model was tested as well as measurement invariance in the latent variables high job demands and low decision authority over time.

Following the mediation analysis guidelines, first a simultaneous equation model that allows for autoregressive and cross-lagged effects between employment in the health and social care industry and a sickness absence of 15 days or more at each time point was estimated. Employment in the health and social care industry was measured at the first time point (t-1, years 2006–2014) and sickness absence at the subsequent time point (t, years 2008–2016). The cross-lagged paths estimated the effect of one variable on the other with a two-year time lag. Each path in the models was adjusted for age, education, civil status, and children living at home. Indicators of the latent variables high job demands and low decision authority were allowed to correlate between waves. The result of this analysis is presented in model 1 below.

In a second step, following the same temporal structure as above, we examined the bivariate multilevel structural cross-lagged relationships between employment in the health and social care industry (predictor variable) and high job demands, low decision authority, and work violence (putative mediators) as well as between high job demands, low decision authority, and workplace violence and sickness absence (outcome). The models were adjusted for the same set of covariates as above. If there are significant paths between the predictor and the mediator and between the mediator and the outcome, a multilevel SEM mediation model can be fitted.

The third step was to apply such a model to our data. A multilevel SEM mediation model (MSEM), in which employment in the health and social care industry was measured at t-2 (in the years 2006–2012), high job demands, low decision authority, and workplace violence at t-1 (in the years 2008–2014) and sickness absence at t (in the years 2010–2016), was fitted. Autoregressive effects as well as cross-lagged paths were estimated between employment in the health and social care industry and the psychosocial work factors and between the psychosocial work factors and sickness absence. These models were adjusted for the same set of covariates as in the bivariate models. The direct effect (the part of the exposure effect that was not mediated through high job demands, low decision authority, or workplace violence) as well as the indirect effect (the part of the exposure effect that was mediated through high job demands, low decision authority, workplace violence) between employment in the health and social care industry and sickness absence were estimated. The results of these analyses are presented in Models 2–4 below.

The multilevel SEM models were built in MPLUS 7. All variables were treated as time-varying variables. Standardised estimates were reported for the final models. The fit statistics chi-square (df), the root mean square error of approximation (RMSEA), the comparative fit index (CFI), the Tucker–Lewis index (TLI), and the standardized root mean square residual (SRMR) were considered. Model fit is assumed to be acceptable when RMSEA ≤ 0.08, TLI ≥ 0.90, CFI ≥ 0.90, and SRMR ≤ 0.08 [42].

## 3. Results

### 3.1. Descriptive Statistics

The distribution of the study variables measured in 2006 are presented in means/standard deviations or n/percentages in Table 1.

### 3.2. The Association between Employment in the Health and Social Care Industry and Sickness Absence (Model 1)

Employment in the health and social care industry, compared with other industries on the Swedish labour market, was associated with more register-based sickness absence of 15 days or more two years later (0.032, *p* = 0.002). As shown in the path from health and social care industry t-1 and health and social care industry t (0.918, *p* = 0.000, see Table 2), few participants changed employment industry over these years. There was no indication of a selection of individuals with more sickness absence in the health and social care industry (0.005, *p* = 0.225). The fit statistics of the model were RMSEA 0.000, CFI 1.000, TLI 1.000, and SRMR 0.000.

Employment in the health and social care industry was positively associated with low decision authority (0.161, *p* = 0.000) and exposure to workplace violence (0.252, *p* = 0.000). High job demands (0.030, *p* = 0.003), low decision authority (0.036, *p* = 0.043), and exposure to workplace violence (0.037, *p* = 0.000), on the one hand, were furthermore associated with a sickness absence of 15 days or more two years later, on the other.

### 3.3. The Mediating Role of High Job Demands (Model 2)

In the mediation model of high job demands, an association between high job demands and a sickness absence 15 days or more (0.031, *p* = 0.003), and an association between employment in the health and social care industry and a sickness absence of 15 days or more (0.030, *p* = 0.002) were observed (see Figure 1). There was no indication of an indirect effect of high job demands in the association between employment in the health and social care industry and sickness absence (0.001, *p* = 0.209). Fit statistics RMSEA 0.036, CFI 0.951, TLI 0.918, and SRMR 0.028.c

### 3.4. The Mediating Role of Low Decision Authority (Model 3)

In the model estimating the mediating role of low decision authority (see Figure 2), we observed an association between employment in the health and social care industry and low decision authority (0.086, *p* = 0.000), between low decision authority and a sickness absence of 15 days or more (0.029, *p* = 0.002), and a direct effect between employment in the health and social care industry and sickness absence (0.028, *p* = 0.005). A small indirect effect of low decision authority in the association between employment in the health and social care industry and a sickness absence of 15 days or more was observed (0.002, *p* = 0.006). The fit statistics of the model were RMSEA 0.030, CFI 0.995, TLI 0.946, and SRMR 0.008.

### 3.5. The Mediating Role of Exposure to Workplace Violence (Model 4)

As shown in Figure 3, employment in the health and social care industry was positively associated with exposure to workplace violence (0.256, *p* = 0.000) and sickness absence (0.022, *p* = 0.036). Additionally, workplace violence predicted a sickness absence of 15 days or more (0.031, *p* = 0.002). A small indirect effect of exposure to workplace violence (0.008, *p* = 0.002) in the association between employment in the health and social care industry and sickness absence was observed. Fit statistics were RMSEA 0.059, CFI 0.973, TLI 0.678, and SRMR 0.012.

## 4. Discussion

The main finding of the present study was that low decision authority and exposure to workplace violence partially mediated the association between employment in the health and social care industry and a register-based sickness absence of 15 days or more. It should be noted, however, that the larger part of the association was not explained by the factors investigated in the present study.

### 4.1. The Association between Employment in the Health and Social Care Industry and Sickness Absence

The result of this study confirms Swedish national statistics of high sickness absence rates in the health and social care industry compared with other industries [1] and does not support the notion that individuals with high sickness absence rates are selected into the health and social care industry from other industries. The rather modest association between employment in the health and social care industry and register-based sickness absence may be due to other industries that are in the reference group in the analyses, such as the education and the building industry, which also have high sickness absence rates in Sweden [1]. Because the aim of the study was to investigate if the established poorer psychosocial working conditions in the health and social care industry compared to other industries could explain the high all-cause sickness absence rates in this industry, we found the alternative method of putting all other industries in the reference category to be the most reasonable solution.

### 4.2. The Mediating Role of High Job Demands

High demands were not found to be more common in the health and social care industry than in other industries, which may not be surprising given that the highest demands on the Swedish labour market have been found in the educational industry [7]. As expected, and in line with previous research [16,17,19,20], there was an association between high job demands and sickness absence. The reverse association was not statistically significant, indicating that individuals that have been on a sickness absence of 15 days or more do not perceive a difference in the level of job demands two years later. This study gives, to conclude, no support that higher sickness absence rates in the health and social care industry are due to higher demands in this industry compared with other industries.

### 4.3. The Mediating Role of Low Decision Authority

The results support previous studies indicating higher levels of low decision authority in the health and social care industry as compared with all other industries on the Swedish labour market [7]. As shown in some previous studies [15,16,18,19], but not others [17], there was also an association between low decision authority and sickness absence. Interestingly, there was additionally a link between sickness absence and lower decision authority at work two years later. This could be due to the fact that individuals with previous sickness absence are more likely to end up in poorer job situations or that individuals with previous sickness absence perceive their situation as having lower decision authority due to their condition. There is a small mediating effect of low decision authority in the association between employment in the health and social care industry and sickness absence, indicating that this dimension of the work environment is relevant for our understanding of sickness absence risks in the health and social care industry.

### 4.4. The Mediating Role of Exposure to Workplace Violence

The results support a large amount of previous studies showing that the health and social care industry is considerably more exposed to workplace violence than other industries [43], and that exposure to workplace violence is associated with a higher risk of sickness absence [30]. A reverse path was also observed in this model, such that individuals with a sickness absence of 15 days or more reported more violence two years later. To what extent this reflects a changed work situation, or a changed perception of the work situation, needs to be illuminated in future research. Exposure to workplace violence did, to a small extent, mediate the association between employment in the health and social care industry and sickness absence, lending further evidence that this is an important part of the work environment to address to decrease sickness absence rates in the health and social care industry.

This study presents evidence that industry level is relevant for our understanding of sickness absence risks due to the psychosocial work environment. Although the occupational level has been found to better distinguish between sickness absence risks in Sweden [1], the organizational prerequisites for a good psychosocial work environment is often established at the industry or workplace level, which is why these levels must be considered important from an intervention perspective. Low decision authority and exposure to workplace violence appear to be factors associated with increased risk of sickness absence for employees on the industry level, indicating that the healthcare industry at large should address these issues to decrease sickness absences rates. To the best of our knowledge, there are no previous studies comparing and explaining differences in sickness absence rates on the industry level.

## 5. Strengths and Limitations

To the best of our knowledge, the present study is among the first to investigate the longitudinal associations between employment in the health and social care industry and sickness absence under the consideration of mediation by psychosocial work factors. Another strength of the study is the use of multilevel SEM models. These models combine the advantages of multilevel modelling and structural equation modelling, i.e., they take into consideration the longitudinal nature of the data (repeated measures nested within individuals) and at the same time allow the study of complex relationships among latent variables [44,45]. Further, this method allows correction for sampling and measurement error, as well as the examination of direct and indirect effects at each level [38]. As recommended in mediation modelling literature, we included measurements at three time-points to appropriately account for the temporal ordering between the three measures. Finally, we measured bidirectional associations while allowing correlations between all constructs and the errors of individual items over time to account for consistency in item-specific variance, which contribute to our understanding of causality [34].

A limitation of the study is that a time lag of two years may not be optimal to capture the development between, for example, workplace factors and sickness absences. Furthermore, sickness absences that are 14 days or less are not captured by Swedish registers. Employees may have several shorter spells of sickness absence over a year due to poor psychosocial working conditions that were not captured by the measure of sickness absence used in this study. Thus, it is unclear if the results of the study can be generalised to shorter spells of sickness absence.

## 6. Conclusions

The present study, based on a representative sample of the Swedish working population using register data to ascertain employment industry and sickness absence and using sound statistical methodology, lends support to the idea that psychosocial work factors are relevant for understanding the increased risks of sickness absence in the health and social care industry.

## Figures and Tables

**Figure 1 ijerph-19-00053-f001:**
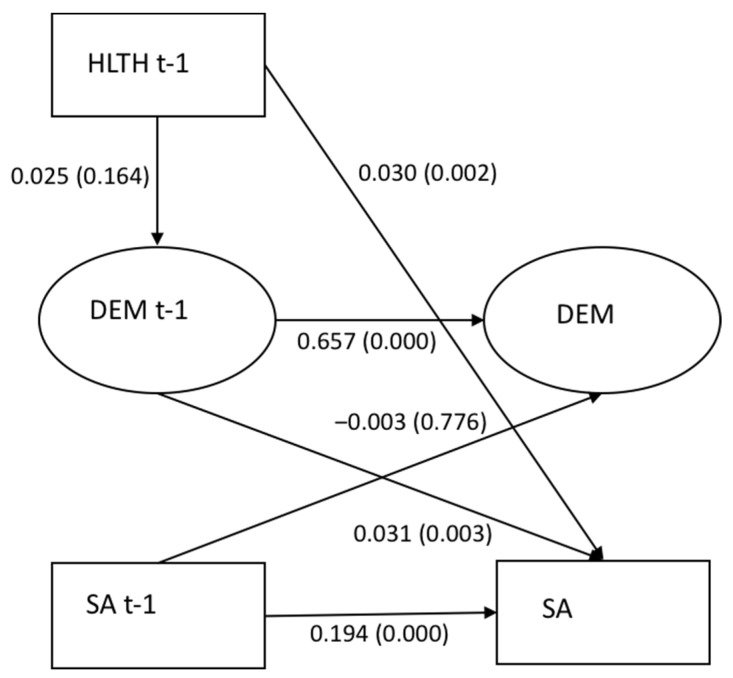
(Model 2). Mediation of high job demands in the association between employment in the health and social care industry (HLTH) and a sickness absence of 15 days or more (SA).

**Figure 2 ijerph-19-00053-f002:**
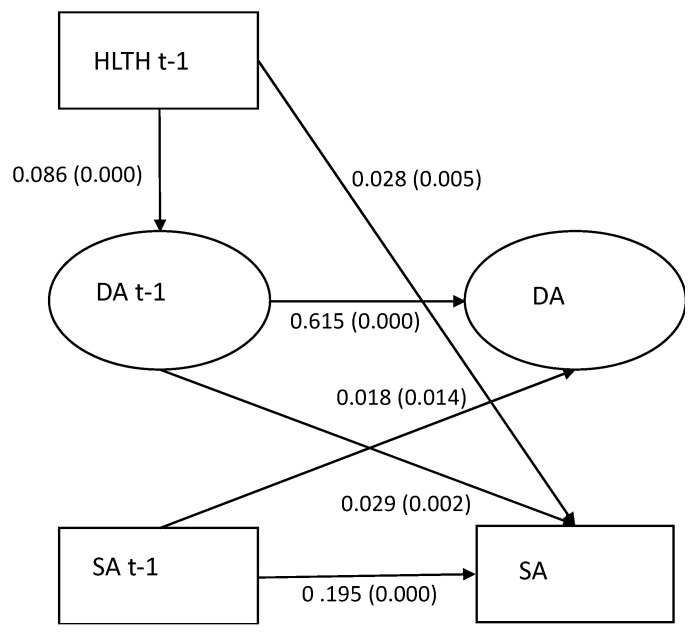
(Model 3). Mediation of low decision authority (DA) between employment in the health and social care industry (HLTH) and a sickness absence of 15 days or more (SA).

**Figure 3 ijerph-19-00053-f003:**
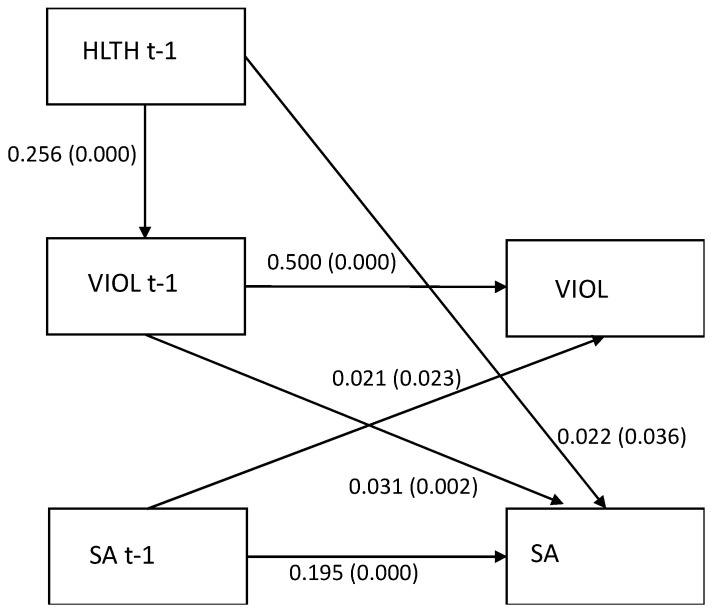
(Model 4). Mediation of workplace violence (VIOL) between employment in the health and social care industry (HLTH) and a sickness absence of 15 days or more (SA).

**Table 1 ijerph-19-00053-t001:** Descriptive statistics for men and women separately, and total sample, Swedish Longitudinal Occupational Survey of Health, 2006.

	Women		Men		Total	
	n/means	%/st.dev	n/means	%/st.dev	n/means	%/st.dev
*Predictor*						
Employment in Health and Social Care						
Yes	675	32.05	105	5.80	780	19.92
No	1431	67.95	1705	94.20	3136	80.08
*Putative mediators*						
Job Demands	2.59	0.56	2.58	0.53	2.59	0.55
Decision Authority	1.72	0.48	1.68	0.49	1.70	0.49
Workplace Violence						
No	1626	77.06	1601	88.36	3227	82.28
Yes	484	22.94	211	11.64	695	17.75
*Outcome variable*						
Sickness Absence						
0 days	1801	84.79	1669	91.45	3470	87.87
>0 days	323	15.21	156	8.55	479	12.13
*Demographic characteristics*						
Gender						
Women	-	-	-	-	2126	53.78
Men	-	-	-	-	1826	46.22
Age						
1 < 34 years	249	11.94	225	12.53	474	12.21
2 35–44 years	522	25.04	470	26.17	992	25.56
3 45–54 years	729	34.96	542	30.18	1271	32.75
4 55–64 years	580	27.82	537	29.90	1117	28.78
5 > 64 years	5	0.24	22	1.22	27	0.70
Education						
1 ≤ 9 years	303	14.35	324	17.94	627	16
2 ≤ 12 years	431	20.45	454	25.14	885	22.59
3 University < 3 years	368	17.42	407	22.54	775	19.78
4 University ≥ 3years	376	17.80	197	10.91	573	14.62
5 Research Education	634	30.02	424	23.48	1058	27
Marital Status						
0 Not Married/cohabited	932	43.86	812	44.47	1744	44.14
1 married/cohabited	1193	53.14	1014	55.53	2207	55.86
Children living at home						
0 No	1021	48.48	851	47.49	1872	48.02
1 Yes	1085	51.52	941	52.51	2026	51.98

**Table 2 ijerph-19-00053-t002:** Standardized parameters, standard errors, and *p*-values for models 1–4.

	B (SE)	*p*
1. Health and social care (HLTH)—Sickness Absence (SA) model		
Sickness Absence (SA) t		
Sickness Absence (SA) t-1	0.195 (0.013)	0.000
Health and social care (HLTH) t-1	0.032 (0.010)	0.002
Age	−0.033 (0.010)	0.001
Children living at home	−0.017 (0.010)	0.109
Married/co-habitant	−0.022 (0.009)	0.015
Education	−0.050 (0.009)	0.000
Gender	0.069 (0.008)	0.000
Health and social care (HLTH) t		
Health and social care (HLTH) t-1	0.918 (0.006)	0.000
Sickness Absence (SA) t-1	0.005 (0.004)	0.225
Age	0.003 (0.004)	0.514
Children living at home	−0.003 (0.003)	0.362
Married/co-habitant	−0.002 (0.003)	0.407
Education	0.000 (0.003)	0.935
Gender	0.025 (0.003)	0.000
2. Health and social care (HLTH)—High demands (DEM)—Sickness Absence (SA) model		
Demands (DEM) t-1		
Health and social care (HLTH) t-1	0.025 (0.018)	0.164
Age	−0.060 (0.018)	0.001
Children living at home	0.015 (0.016)	0.360
Married/co-habitant	0.002 (0.016)	1.351
Education	0.137 (0.018)	0.000
Gender	0.068 (0.017)	0.000
Demands (DEM) t		
Sickness Absence (SA) t-1	−0.003 (0.009)	0.776
Demands (DA) t-1	0.657 (0.024)	0.000
Age	−0.060 (0.018)	0.001
Children living at home	0.015 (0.016)	0.360
Married/co-habitant	0.022 (0.016)	0.177
Education	0.137 (0.018)	7.728
Gender	0.068 (0.017)	3.899
Sickness absence (SA) t		
Sickness absence (SA) t-1	0.194 (0.013)	0.000
Health and social care (HLTH) t-1	0.030 (0.010)	0.002
Demands (DEM) t-1	0.031 (0.010)	0.003
Age	−0.032 (0.010)	0.001
Children living at home	−0.017 (0.010)	0.099
Married/co-habitant	−0.023 (0.009)	0.012
Education	−0.054 (0.009)	0.000
Gender	0.067 (0.008)	0.000
3. Health and social care (HLTH)—Low decision authority (DA)—Sickness absence (SA) model		
Decision Authority (DA) t-1		
Health and social care (HLTH) t-1	0.086 (0.014)	0.000
Age	−0.122 (0.014)	0.000
Children living at home	−0.021 (0.014)	0.135
Married/co-habitant	−0.041 (0.014)	0.002
Educ	−0.173 (0.013)	0.000
Sex	0.084 (0.015)	0.000
Decision Authority (DA) t		
Decision Authority (DA) t-1	0.615 (0.009)	0.000
Sickness absence (SA) t-1	0.018 (0.007)	0.014
Age	−0.030 (0.008)	0.000
Children living at home	−0.001 (0.007)	0.924
Married/co-habitant	−0.008 (0.007)	0.231
Education	−0.066 (0.007)	0.000
Gender	0.048 (0.007)	0.000
Sickness absence (SA) t		
Sickness absence (SA) t-1	0.195 (0.013)	0.000
Health and social care (HLTH) t-1	0.028 (0.010)	0.005
Decision Authority (DA) t-1	0.029 (0.009)	0.002
Age	−0.030 (0.010)	0.005
Children living at home	−0.016 (0.010)	0.121
Married/co-habitant	−0.021 (0.009)	0.022
Education	−0.045 (0.009)	0.000
Gender	0.067 (0.008)	0.000
4. Health and social care (HLTH)—Workplace violence (VIOL)—Sickness absence (SA) model		
Workplace violence (VIOL) t-1		
Health and social care (HLTH) t-1	0.256 (0.016)	0.000
Age	−0.112 (0.014)	0.000
Children living at home	−0.001 (0.014)	0.937
Married/co-habitant	−0.022 (0.012)	0.081
Education	−0.011 (0.012)	0.386
Gender	0.039 (0.012)	0.001
Workplace violence (VIOL) t		
Workplace violence (VIOL) t-1	0.500 (0.015)	0.000
Sickness absence (SA) t-1	0.021 (0.009)	0.023
Age	−0.030 (0.010)	0.002
Children living at home	−0.017 (0.010)	0.107
Married/co-habitant	−0.021 (0.009)	0.018
Education	−0.050 (0.009)	0.000
Gender	0.068 (0.008)	0.000
Sickness absence (SA) t		
Sickness absence (SA) t-1	0.195 (0.013)	0.000
Health and social care (HLTH) t-1	0.022 (0.010)	0.036
Workplace violence (VIOL) t-1	0.031 (0.010)	0.002
Age	−0.030(0.010)	0.036
Children living at home	−0.017(0.010)	0.107
Married/co-habitant	−0.021(0.009)	0.018
Education	−0.050(0.009)	0.000
Gender	0.068(0.008)	0.000

## Data Availability

Data from the SLOSH cohort are not publicly available due to legal restrictions. For data requests, please contact data manager Constanze Leineweber at contanze.leineweber@su.se.

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
