# Peer review of "Does Exposure to High Job Demands, Low Decision Authority, or Workplace Violence Mediate the Association between Employment in the Health and Social Care Industry and Register-Based Sickness Absence? A Longitudinal Study of a Swedish Cohort"

_ijerph, 2021, doi:10.3390/ijerph19010053_

Round 1

Reviewer 1 Report

The reviewer appreciated the opportunity to review this manuscript focused on causal mechanisms for sickness absence among employees in the health and social care industry in Sweden. Overall, the research study and manuscript are well-designed. However, the reviewer does have some key suggestions for improving the manuscript prior to it being suitable for publication. The suggestions, by section, are provided below.

Introduction:

The authors need to provide much more literature support for the study. This was the biggest weakness of the manuscript. The authors must do a much better and thorough job of setting up the argument or need for the current study. The reviewer would recommend subsections on the three key aspects of the study: sickness absence, psychosocial working conditions, and workplace violence. 

Methods:

The study design needs more details regarding the SLOSH. It is good that the authors cited the profile, but they still need to provide further details on how the SLOSH was designed, the purpose, etc. 

The reviewer would also suggest more rationale for why the analytical strategy was chosen. The reviewer, in general, agrees with the strategy, but it would be good to provide some rationale for why it was selected.

Results:

Overall, the results are presented well and in an organized fashion (easy to follow).

Discussion/Conclusions:

Overall, the discussion and conclusions are supported well by the results.

Author Response

Reviewer 1

Comments and Suggestions for Authors

The reviewer appreciated the opportunity to review this manuscript focused on causal mechanisms for sickness absence among employees in the health and social care industry in Sweden. Overall, the research study and manuscript are well-designed. However, the reviewer does have some key suggestions for improving the manuscript prior to it being suitable for publication. The suggestions, by section, are p(rovided below.

Introduction:

The authors need to provide much more literature support for the study. This was the biggest weakness of the manuscript. The authors must do a much better and thorough job of setting up the argument or need for the current study. The reviewer would recommend subsections on the three key aspects of the study: sickness absence, psychosocial working conditions, and workplace violence. 

Response: Thank you for this comment and your recommendations, we agree and have now added more literature support and argumentation for the need of the study, see introduction page 1-2.

Methods:

The study design needs more details regarding the SLOSH. It is good that the authors cited the profile, but they still need to provide further details on how the SLOSH was designed, the purpose, etc. 

Response: We have now added the following paragraph, page 3:

The overall aim of the SLOSH survey is to facilitate studies of how psychosocial work factors, the interface between work and family life, and the transition to retirement, among others, affect health outcomes over time. By continuously inviting participants of the Swedish Work Environment Surveys (SWES) to follow-ups, Statistics Sweden have collected data from the SLOSH cohort every second year since 2006. Today SLOSH comprises SWES participants from 2003 until 2011 with a sample size of over 40,000 individuals. SWES consist of a subsample of gainfully employed people aged 16–64, stratified by county, sex, citizenship and inferred employment status from the Labour Force Survey (LFS). SLOSH respondents are invited to answer a self-completion questionnaire in two versions, one for those who work 30 % or more of full time and another one for those who work less than 30% or not at all.

The reviewer would also suggest more rationale for why the analytical strategy was chosen. The reviewer, in general, agrees with the strategy, but it would be good to provide some rationale for why it was selected.

Response: Thank you for this suggestion. We believe that we have given a rationale for why the analytical strategy was chosen by the following sentence and references, page 4:

“The autoregressive cross-lagged models are among the most popular approaches in mediation analysis with longitudinal panel data (38) since they allow time for causes to have their effects, supports stronger inference about the direction of causation in comparison and reduces the probable parameter bias (37).”

However, to follow your suggestion we have also clarified the steps in the analytical strategy in the following section, page 5:

“Bivariate models were first fitted to examine whether there were cross-lagged relationships between the exposure of interest and the putative mediators, and between the putative mediators and outcome, which is a prerequisite for a causal pathway (41). We tested models where high job demands and low decision authority were fitted either as latent variables with four and two items or as observed variables (considering the mean of the various items) and finally presented only the results for the models with the best fit. The fit of the measurement model was tested as well as measurement invariance in the latent variables high job demands and low decision authority over time.

Following the mediation analysis guidelines, first a simultaneous equation model that allows for autoregressive and cross-lagged effects between employment in the health and social care industry and sickness absence 15 days or more at each time point was estimated.”

Results:

Overall, the results are presented well and in an organized fashion (easy to follow).

Response: Thank you.

Discussion/Conclusions:

Overall, the discussion and conclusions are supported well by the results.

Response: Thank you.

Reviewer 2 Report

The authors present an interesting paper however there are some issues that need to be addressed:

  1. The references are set after the period, and should be the other way around.
  2. Some sentences are too long and other two short. This issue is present by the lack of connectors resulting on a lack of flow
  3. The study relevance and objectives are two connected and there is a difficulty to differentiate between both
  4. The authors present clearly the methods but they don't follow the structure of the manuscript

On the positive side, the discussion and conclusion are great, being need little modifications regarding expressions.

Author Response

Reviewer 2

Comments and Suggestions for Authors

The authors present an interesting paper however there are some issues that need to be addressed:

  1. The references are set after the period, and should be the other way around.

Response: Thank you for drawing our attention to this, it has now been changed.

  1. Some sentences are too long and other two short. This issue is present by the lack of connectors resulting on a lack of flow

Response: Thank you, we have now gone through and edited the text to create a better flow in the manuscript (see particularly Introduction and Discussion, strengths and limitations).

  1. The study relevance and objectives are two connected and there is a difficulty to differentiate between both

Response: We have now rewritten the introduction, page 1-2, to address this and other issues raised about the introduction.  

  1. The authors present clearly the methods but they don't follow the structure of the manuscript

Response: Thank you for this comment. We organized the results and discussion sections around the four statistical models that we present and that make up the main results of the study. It is correct that the methods section includes information also on other analyses that we conducted and for which the results were not reported in similar detail. In order to clarify how the analyses in the methods section relate to the results and discussion section, we have now included a sentence with reference to the model in which the results for each analysis can be found, see page 5.

We have also improved the structure of the introduction by inserting headings for exposure, outcome and mediating variables and elaborated on the rationale for the study, see page 1-2.

On the positive side, the discussion and conclusion are great, being need little modifications regarding expressions.

Response: Thank you.

Round 2

Reviewer 1 Report

The authors have sufficiently addressed the reviewer's original suggestions for improvement. The reviewer has no further comments or suggestions.